



# The linkage between autumn Barents-Kara sea ice and European cold winter extremes

Di Cai[1,2], Gerrit Lohmann[1,3], Xianyao Chen[2] and Monica Ionita[1,4,5]

[1] Alfred Wegener Institute Helmholtz Center for Polar and Marine Research, Bremerhaven, Germany

[2] Frontiers Science Center for Deep Ocean Multi-spheres and Earth System and Key Laboratory of Physical Oceanography, Ocean University of China, and Pilot National Laboratory of Marine Science and Technology (Qingdao), Qingdao 266237, China

[3] University of Bremen, Bremen, Germany

[4] Emil Racovita Institute of Speleology, Romanian Academy, Cluj-Napoca, 400006, Romania

[5] Faculty of Forestry, "Stefan cel Mare" University of Suceava, Suceava, Romania

*Correspondence to*: Di Cai (di.cai@awi.de)

**Abstract.** Despite intense efforts to understand the links between the Arctic region and mid-latitudes, there is no consensus on the relationship between sea ice retreat and the frequency of occurrence of mid-latitude weather extremes (e.g., cold spells,

heatwaves, droughts). By tracking monthly variabilities based on observational data, we show that a decline in sea ice over the Barents-Kara Seas in autumn is related to extreme cold winters over much of Europe. The winter temperature change in Europe is a direct response to a stationary Rossby wave generated by the lower troposphere diabatic heat anomaly as a result of sea ice loss over the Barents-Kara Seas in autumn, leading to a negative phase of North Atlantic Oscillation and more frequent episodes of the atmospheric blocking over Greenland and the North Atlantic. The negative phase of the North Atlantic

Oscillation and enhanced blocking are closely related and mutually reinforcing, shaping the spatial distribution of cold anomalies over much of the European continent. Our results suggest a link between the unusual retreat in Barents-Kara Sea ice during autumn and the occurrence of intense European weather extremes in subsequent winter months. Nevertheless, climate models have difficulties to capture the variability and trend of the Artic sea ice and to capture the relationship between sea ice reduction and European winter extremes. Consequently, further work on this relationship on monthly time scales will

improve our understanding of the prediction of midlatitude extreme events.



## 1 Introduction

Observations indicate a significant warming of the Arctic region over the recent decades, accompanied by the accelerating decline of sea ice cover in all regions and for all seasons (Screen and Simmonds, 2010; Stroeve et al., 2011; Stroeve and Notz, 2018; Ionita, 2023). The linear trend of the Arctic Sea ice extent in September is about -13% per decade during 1979-2017 (Serreze and Meier, 2019). These variations in the Arctic sea-ice conditions may trigger local and remote concurrent processes, influencing the regional surface energy budget (Screen and Simmonds, 2010), atmospheric and ocean circulation patterns (Francis et al., 2009; Sévellec et al., 2017), and weather systems over Northern Hemisphere continents (Cohen et al., 2014).

Despite intense efforts to understand Arctic-midlatitudes linkages, the controversy remains regarding a causal relationship between the sea ice reduction and mid-latitude weather extremes, especially at monthly scale. Several studies have used numerical models to assess their connections by forcing the climate models with reduced sea ice conditions and comparing the corresponding midlatitude temperature responses to those from simulations with increased sea ice extent, but the results do not support a robust causation (Sun et al., 2016; McCusker et al., 2016). Blackport et al. (2019) conclude that mid-latitude cooling in winter is not caused by reduced sea ice, but rather by atmospheric circulation changes that precede and then simultaneously drive cold mid-latitude winters and mild Arctic conditions. In contrast, several studies based on observations (Hopsch et al., 2012; Tang et al., 2013) and model simulations (Mori et al., 2014) do find a robust connection between cold winter extremes over the northern hemisphere and the Arctic sea-ice loss, suggesting that the latter forces the former.

Thermodynamic forcing by sea ice reduction contributes to weather events by changing the large-scale atmospheric circulation and the internal shifts in atmospheric dynamics (Overland et al., 2021). More specifically, the reduction of Arctic Sea ice in autumn, particularly over the Barents and Kara Seas (BKS) (Petoukhov and Semenov, 2010), can lead to warming and instability of the overlying atmosphere. The resulting preferential warming of the Arctic atmospheric column further causes an increase in geopotential height thickness and a decrease in the meridional temperature gradient, which may enhance upward propagating planetary waves into the stratosphere (Honda et al., 2009) and slow the polar jet stream (Francis et al., 2009).

On one hand, sufficient wave breaking in the stratosphere is conducive to disrupting and weakening the stratospheric polar vortex (SPV), potentially triggering stratospheric warming events (Cohen et al., 2014). Weeks to months later, the circulation feature of the weakened polar vortex descends into the troposphere and surface, evident as a negative phase of the North Atlantic Oscillation (NAO) and associated continental cooling conditions (Jaiser et al., 2013; Wegmann et al., 2020), implying an essential role for tropospheric-stratospheric coupling (Sun et al., 2016). On the other hand, weakening zonal winds increase the likelihood of slower and more amplified Rossby waves, leading to more frequent blocking (Francis and Vavrus, 2012; Preece et al., 2023) characterized by persistent anticyclones disrupting the steady westerly flow. Due to its large spatial extent and temporal persistence, atmospheric blocking may cause large-scale circulation anomalies that substantially impact weather patterns and are often associated with significant climate anomalies (Häkkinen et al., 2011; Rimbu and Lohmann 2011; Ionita et al., 2016).



60       However, what remains unclear is how the monthly atmospheric circulation and the frequency of cold spells respond to the reduced autumn sea ice in the BKS. The capability of climate models to simulate atmospheric circulations, the frequency and location of blocking (Masato et al., 2013), and winter temperatures (Cohen et al., 2020) is limited, thus inhibiting the understanding of the physical processes involved in the response of extreme cold weather to sea ice reductions through atmospheric changes. Moreover, various extreme cold winters have occurred in the mid-latitude continents during ongoing

global warming and sea ice decline (Cohen et al., 2014; Kretschmer et al., 2018), affecting individuals, agriculture, and commerce directly. In this respect, here we trace the development of atmospheric flow and the blocking frequency on the interannual timescale, to present observational evidence that a reduced sea ice extent (SIE) is affecting European extreme winter weather. In brief, the aim of this study is to answer three questions: (1) Is there a link between autumn sea ice reduction and monthly extreme cold events in Europe? If so, (2) by what physical process does it affect the temperature anomaly? And

further to explore (3) the time evolution of atmospheric conditions in response to sea ice loss on the interannul timescale.

## 2 Data and Methods

### 2.1 Data

      Observations of monthly SIE time series in the Barents-Kara Seas, in autumn (SON) for the period 1979-2020, were obtained from National Snow and Ice Data Center (NSIDC, Fetterer et al., 2017). To investigate the large-scale atmospheric

circulation in response to the initial sea ice forcing, we used the monthly sea level pressure (SLP), the 2m temperature (T2M), geopotential height and zonal and meridional winds at 500 and 100 hPa, with a grid-point resolution of 0.25°×0.25°, in winter (DJF) from 1979/1980 to 2020/2021, which are available from the fifth generation European Centre of Medium-Range Weather Forecasts atmospheric reanalysis (ERA5, Hersbach et al., 2020). Before the analysis, all variables were detrended in order to eliminate the effects of global warming and accentuate the fluctuations.

80       We further used monthly outputs from 13 CMIP6 models (Eyring et al., 2016; Table 1) to assess the performance of climate models to simulate the interannual variability of sea ice and the large-sale atmospheric circulation. For each model, the historical (1979-2014) and SSP5-8.5 (2015-2021) simulations from the r1i1p1f1 ensemble member were concatenated to produce a continuous time record in this study. To facilitate calculations and comparisons, we re-gridded all these data to a uniform spatial resolution (1°×1°) via a bilinear interpolation.


**Table 1**. List of CMIP6 models used in this study.

| | CMIP6 Model Name | Horizontal resolution (lon. by lat. in degrees) |
|---|---|---|
| 1 | ACCESS-CM2 | 1.9°×1.3° |
| 2 | ACCESS-ESM1-5 | 1.9°×1.2° |



| 3 | BCC-CSM2-MR | 1.1°×1.1° |
|---|---|---|
| 4 | CAMS-CSM1-0 | 1.1°×1.1° |
| 5 | CanESM5 | 2.8°×2.8° |
| 6 | CMCC-CM2-SR5 | 1°×1° |
| 7 | CMCC-ESM2 | 0.9°×1.3° |
| 8 | FGOALS-g3 | 2°×2.3° |
| 9 | FIO-ESM-2-0 | 1.3°×0.9° |
| 10 | MIROC6 | 1.4°×1.4° |
| 11 | MPI-ESM1-2-LR | 1.9°×1.9° |
| 12 | MRI-ESM2-0 | 1.1°×1.1° |
| 13 | NorESM2-LM | 2.5°×1.9° |

## 2.2 Indices for extreme climate events

In this study, extreme climate events over Europe are captured by the monthly temperature index, TN10p, as defined by the Expert Team on Climate Change Detection and Indices (ETCCDI, Peterson et al., 2001). TN10p is an index measuring the percentage of days when the daily minimum temperature is below the 10th percentile threshold calculated for each calendar day (regarding the climatological norm) using a running 5-day window. This is a measure of the percentage of cold nights in winter. The observational daily minimum temperature (TN) has been extracted from the E-OBS database v23.1e (Cornes et al., 2018) with a spatial resolution of 0.1°×0.1°.

## 2.3 Computation of the 2D blocking frequency

We used the two-dimensional (2D) blocking frequency index (Scherrer et al., 2006) to investigate the impacts of the SIE anomalies in the previous autumn on the temperature conditions over Europe in the following winter. The daily Z500 from the ERA5 reanalysis data (Hersbach et al., 2020) during 1979-2020 was used to calculate the blocking frequency. The 2D blocking index is based on the evaluation of blocking conditions at every grid point and goes beyond the one-dimensional blocking index (Tibaldi and Molteni, 1990). This index is obtained in terms of the following geopotential height gradients in the south (GHGS) and north (GHGN) sides of the blocking region at given three reference latitudes:

$$GHGS = \left(Z(\varphi_0) - Z(\varphi_0 - 15°)\right)/15°$$

$$GHGN = \left(Z(\varphi_0 + 15°) - Z(\varphi_0)\right)/15°$$

where $\varphi_0$ is the latitude of the considered grid point.

A grid point is defined as instantaneous blocking if the conditions GHGS > 0 and GHGN < (−10 m/degrees of latitude) are simultaneously satisfied. A blocking event is detected if continuous instantaneous blocking covers at least five consecutive days. For each winter, the blocking frequency is defined as the ratio between the total number of days of blocking events and





the total number of winter days. According to Scherrer et al. (2006), blockings are detected separately at each grid point within the latitude zone 35°-75°N.

## 2.4 Composite analysis

Composite analyses were applied to depict the typical pattern and physical mechanism responsible for connecting between the autumn sea ice extent (SIE) in the Barents-Kara Seas and the atmospheric circulation, surface air temperature, and blocking frequency. We regarded years when the value of the time series of detrended SIE in autumn was above (below) 0.8 standard deviation as high (low) SIE years. To examine the relationship between autumn sea ice and the following European weather conditions, monthly blocking frequency and TN10p anomalies in the subsequent winter-to-autumn were also performed in
composite analysis.

    The threshold value used for the composite analysis does not significantly alter the results. The statistical significance of the composite maps at the 5% significance level is established by Welch's *t*-test (Welch, 1947).

## 3 Results

### 3.1 Analysis of Observations

The linear detrended time series of autumn SIE over the BKS for 1979-2021 (Fig. 1) exhibits substantial interannual variability. To understand the influence of atmospheric circulation responses to different autumn sea-ice states on extreme cold events in Europe, we constructed the composite maps between the linear detrended time series of autumn SIE over the BKS for the years when the values of the index were higher than 0.8 standard deviation (high) and lower than -0.8 standard deviation (low). Statistically, the total number of high and low autumn SIE episodes considered were the same (8 cases), as derived from
Figure 1.

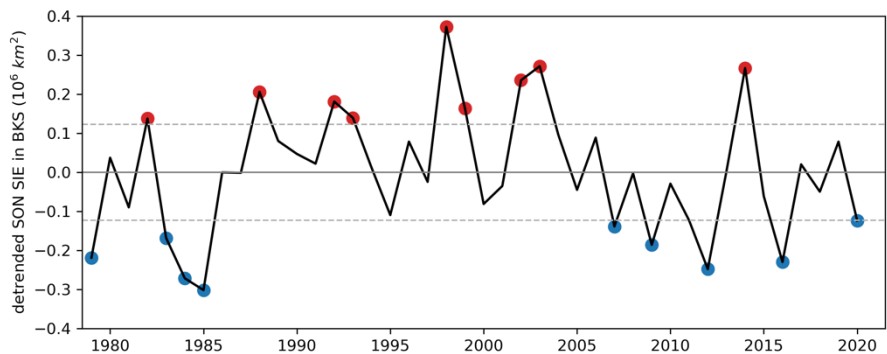



**Figure 1.** The linear detrended autumn SIE over the BKS from 1979 to 2021 in NSIDC observations (Fetterer et al., 2017). Years when the SIE was above and below 0.8 standard deviation are indicated by red and blue dots, respectively.


The wintertime atmosphere anomalies in response to different autumn sea-ice conditions over the BKS are shown in Fig. 2 and 3. In the low SIE case, the composite map of the 100 hPa geopotential height (Z100) anomaly (Fig. 2b) exhibits a wave-like pattern. Strong anomalous easterlies dominate the circulation in response to SIE decrease over the BKS for almost the whole 50°N-70°N latitudinal band, thereby reducing westerly flow. The wind differences match the associated changes of

Z100 field well, with a maximum anomaly of 100 gpm over the central Arctic. Additionally, the spatial characteristics of Z100 (Fig. 2b) and the 500 hPa geopotential height (Z500, Fig. 2d) suggest a weakened polar vortex from the surface to the upper troposphere. The Z500 composite (Fig. 2d) presents concentrated positive values between 90°W and 60°E, with the maxima located in the Greenland Sea. The most crucial feature corresponds to the intensification of the meridional component of the mid-troposphere circulation that is perfectly visible up and downstream of the Greenland Sea during low SIE situations.


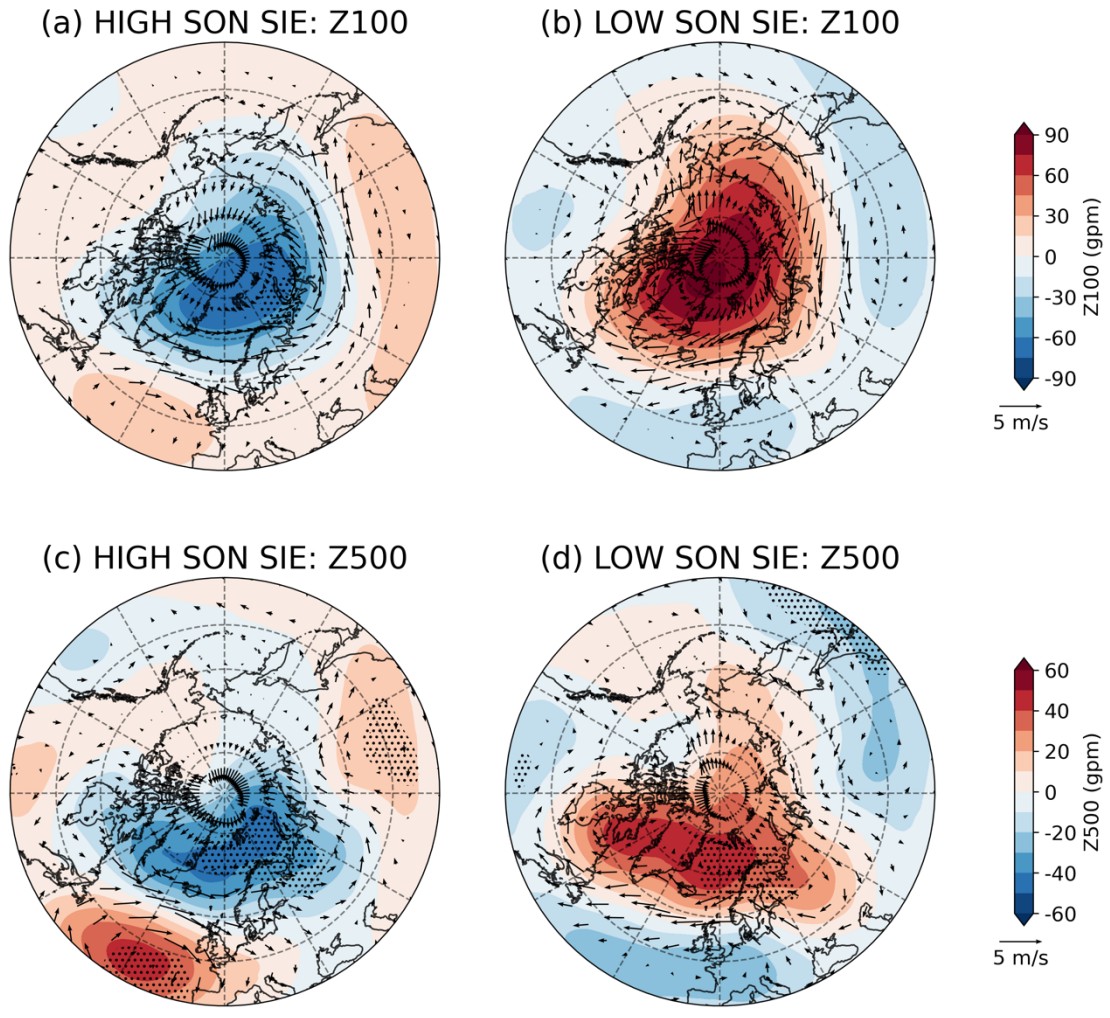

**Figure 2.** Composite maps for the Z100 (a, b), Z500 (c, d) and corresponding wind vectors for winter (DJF) relative to the detrended autumn SIE in BKS above (left) and below (right) 0.8 standard deviation. The dots highlight significant anomalies at a confidence level of 95%.






The SLP pattern (Fig. 3b) resembles the fingerprint of the negative phase of Arctic Oscillation (AO) (Thompson and Wallace, 1998), corresponding to a positive pressure anomaly in the polar region. This structure is caused by the decrease of autumn sea ice in the BKS, which leads to the increase of SST over the BKS region and the strengthening of the Siberian High in winter. Weak zonal winds further result in more cold Arctic air penetrating the mid-latitudes. As such, there is significant
cooling over the mid and high latitudes of the Eurasian continent, as shown in Figure 3d, and the cooling extends southward to the whole of Europe. This pattern appears to be a "warm Arctic and cold continent" pattern (Zhang et al., 2008). Strong warming over the BKS is contrasted by considerable cooling in adjacent northern Eurasia in the following winter. Although of smaller magnitude, negative SAT anomalies also cover a large part of North America.

For the high SIE years, the combination of the strengthened polar vortex (Fig. 2a) and positive AO (Fig. 3a), which favors
the mid-latitude jet stream to blow strongly and consistently from west to east, causes the cold Arctic air to be locked in the polar region, coinciding with the Eurasian warming (Fig. 3c). Notably, a comparison of the wind changes and SAT reveals a good correspondence between temperature and zonal wind changes over Europe in winter, shown in Figure 2 and 3. Anomalous south-westerlies, bringing warm Atlantic air in high SIE situations, are contrasted by anomalous solid north-easterly flow over this region for low SIE cases (Petoukov and Semenov, 2010). The latter is also a distinctive feature of the extreme European
winter 2005-2006 event accompanied by the anomalous north-easterly and strong cooling over the European site, according to NCEP/NCAR reanalysis data (Kalnay et al., 1996).



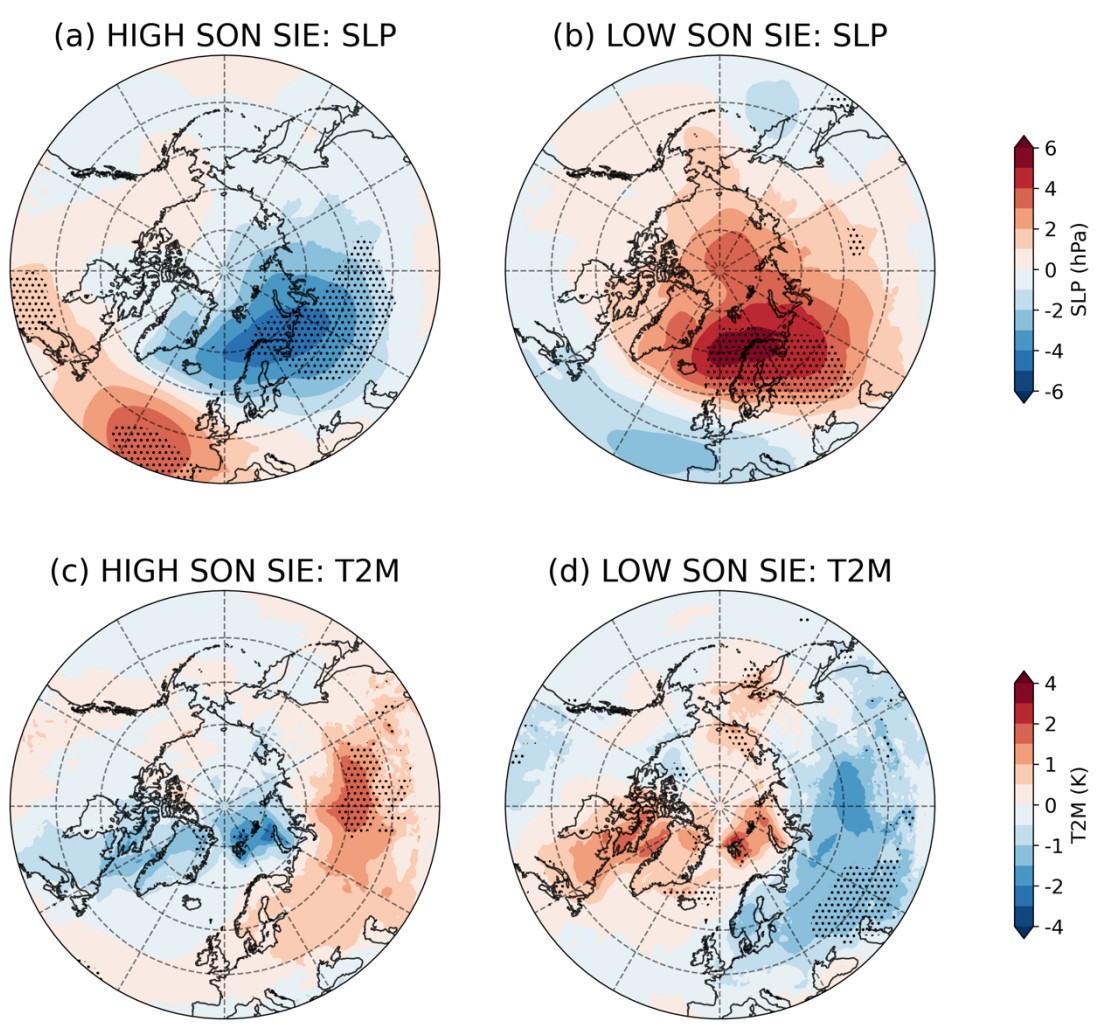

**Figure 3.** Composites of SLP (a, b) and T2M (c, d) for winter (DJF) relative to the detrended autumn SIE in BKS above (left) and below (right) 0.8 standard deviation. The dots highlight significant anomalies at a confidence level of 95%.





To highlight the impact of sea ice reduction on the atmospheric climate regimes, we illustrate the composite maps of Z100 (Fig. 4) as well as Z500 (Fig. 5) and blocking frequency (Fig. 6) anomalies for the following December to May, associated with anomalously low autumn SIE over the BKS. The primary reason for using monthly averages rather than short event durations is that we are interested in persistent conditions on interannual time scales.







**Figure 4.** Composites of Z100 and 100 hPa wind vector anomalies for the following December to May, relative to the detrended autumn SIE in BKS below 0.8 standard deviation (low SON SIE). The dots highlight significant anomalies at a confidence level of 95%.

A major controlling factor for the classic winter cold-air outbreaks into mid-latitudes is the occurrence of SPV disruptions and displacements over the continents. This atmospheric response can be regarded as resulting from the stationary Rossby wave generated by the warming of the lower troposphere, which is caused by the decrease in autumn sea ice cover (Inoue et al., 2012). To better illustrate the factors contributing to SPV disruptions, Figure 4 shows the monthly evolution of the composite Z100 anomaly for the following December to May next year. The choice of such a low stratospheric level depends on this being particularly crucial for the troposphere-stratosphere coupling (Garfinkel et al., 2017).

The positive anomalies dominated the Arctic region for most of the study period, indicating a weakening of the polar vortex associated with stratospheric warming. However, the behavior of SPV in early winter (December) differs from that in mid-winter and spring. In December (Fig. 4a), the vortex may be pushed away from the pole or split. The progressively weaker states of the SPV can be clearly seen from January (Fig. 4b) to March (Fig. 4d), mainly manifested as a positive Z100 anomaly centered over Greenland/Central Arctic. Climatologically, the SPV emerges in the autumn, reaches its strongest state in January, and gradually diminishes in the spring (Zhang et al., 2022). The geopotential height, on the contrary, remains weakened after January. The positive anomalies in February (Fig. 4c) are broader than in January (Fig. 4b), centered on the Davis Strait, and the pattern remains robust in March (Fig. 4d), amplifying the positive anomaly. This pattern is not perfectly symmetrical around the North Pole, but anticyclonic anomalies are displaced toward the western hemisphere. Besides, when examining the temporal evolution of their intensities, we found that Z100 anomalies were at their most robust state in March and then gradually weakened at a slow pace in the following two months. In April (Fig. 4e), it exhibits an east-west contrast over the Arctic region, when Z100 anomalies are divided, having a significant positive center over Russia and a negative center over Canada.



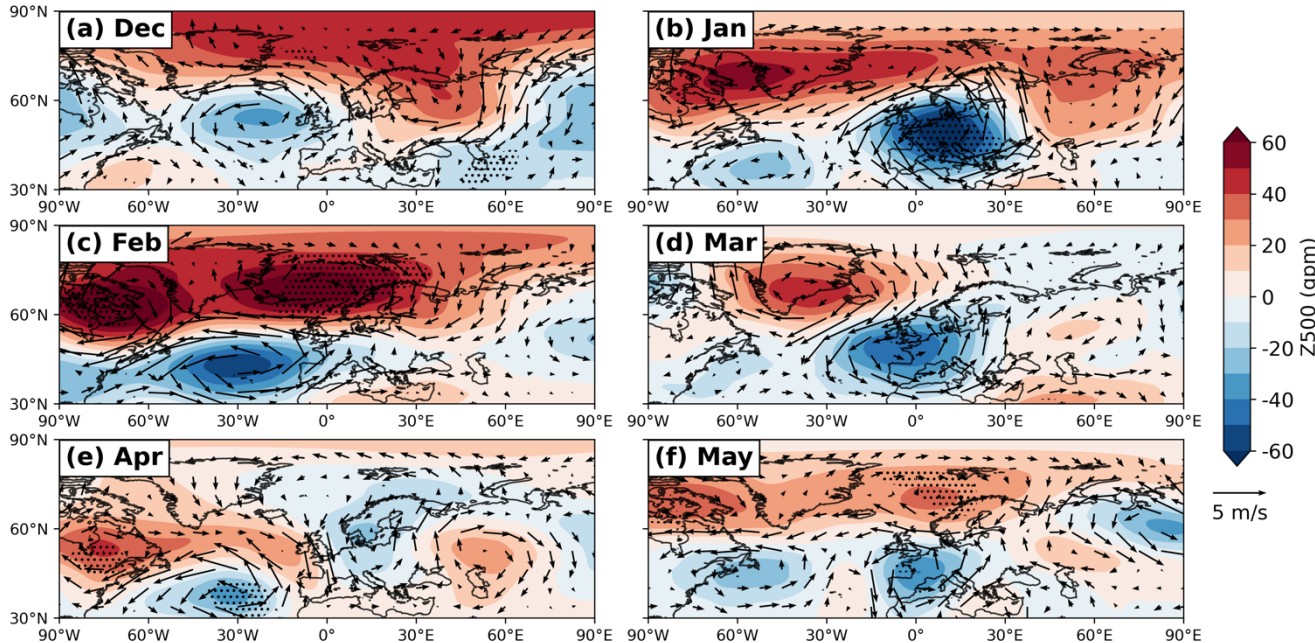

**Figure 5.** Composites of Z500 and 500 hPa wind vector anomalies for the following December to May, relative to the detrended autumn SIE in BKS below 0.8 standard deviation (low SON SIE). The dots highlight significant anomalies at a confidence level of 95%.

Figure 5 tracks the monthly response of Z500 anomaly composite to autumn sea ice reduction, which clearly illustrates the importance of the troposphere-stratosphere coupling to the surface climate (Scaife et al., 2008). The composite map of the Z500 anomalies exhibit a higher degree of zonal symmetry in the lower stratosphere (Fig. 4b-d), but wavy signatures dominate the troposphere with little zonal symmetry. The coherence of the positive and negative anomalies in the troposphere and the lower stratosphere suggests a downward influence by anomalous behavior of SPV on the tropospheric circulation from December to January, consistent with previous findings (e.g., Polvani and Kushner 2002; Kushner and Polvani 2004). In addition, the most crucial feature of the Z500 anomalies corresponds to the dramatical intensification of the meridional component of the mid-troposphere circulation, which is perfectly visible in the following months (Fig. 5). On the contrary, high SIE episodes are characterized by an intensification of the zonal circulation (Fig. 2c).



When tracking the month-by-month evolutions, we readily find pronounced north-south meanders in December (Fig. 5a), with negative anomalies in the mid-latitudes except over the northern Europe, which provided a unique dynamic setting driving cold polar air to plunge southward into Europe. In January (Fig. 5b), the pattern of Z500 anomalies begin to form prototypes, which shows some resemblance to the negative phase of the winter AO, and the jet stream gradually intensifies. Two intense

anticyclonic circulations develop around the west and east coasts of Greenland, while a negative Z500 anomaly can be observed over the southwestern Europe. This pattern expands slightly to the northwest, even magnifying the positive anomalies, and reducing the negative ones, reaching its strongest state in February (Fig. 5c). The latter diminishes and shifts towards the Atlantic, which is relevant for the heat transport of the southern European. This leads to an enforced north-south geopotential height gradient, and thus the atmospheric flow is diverted around it with an intense flow from the north, allowing cold air to

penetrate southwards. In March (Fig. 5d), the anticyclonic vortex decreases over southern Greenland, being accompanied by relatively negative Z500 anomalies. The anomalies associated with the south of the jet location are very similar to the negative pattern of the NAO, and the jet stream is located further south than its climatology. Specifically, a relatively elongated negative anomaly to the south indicates an increased meridional gradient and a strong jet stream lying to the south of the anticyclonic anomaly. As shown in Figure 5b-d, the weakened SPV is transmitted down into the troposphere during late winter, causing

circulation anomalies similar to the negative phase of NAO. As such, the anomalous propagation of the Rossby waves may be the precursor to forcing the NAO signals in the troposphere. In addition, when there is a block over southern Greenland (Fig. 6d) with a negative NAO regime (Fig. 5d) in March, this will force the jet stream to proceed more or less zonally across the Atlantic in April (Fig. 5e), similar to the result from Woollings et al., (2010).





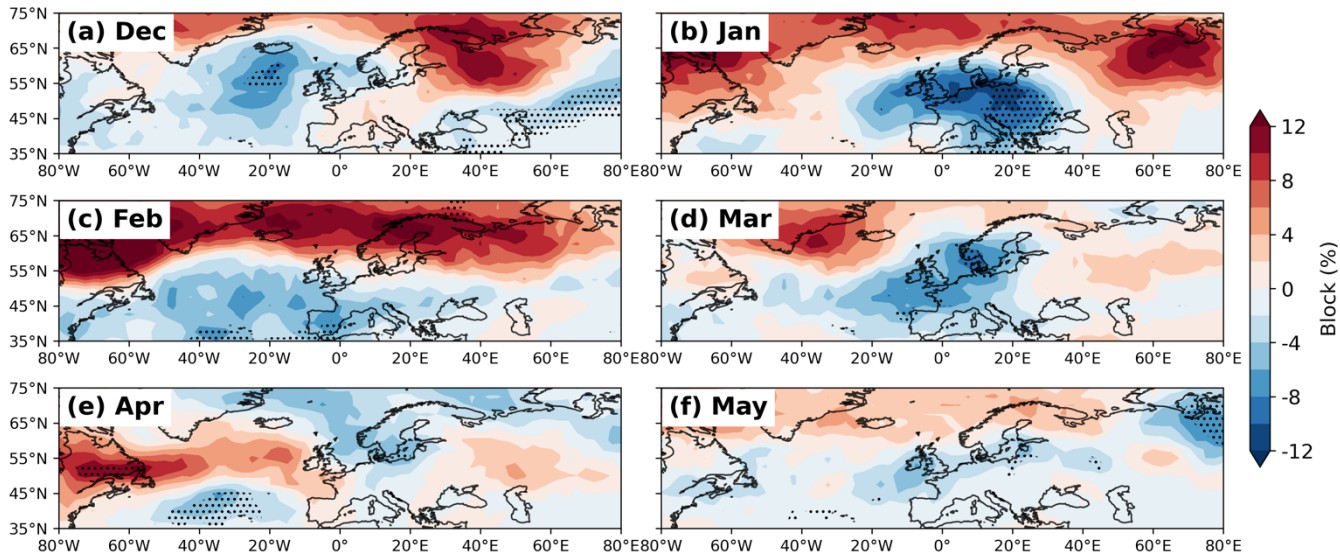

**Figure 6.** Blocking frequency composite maps for the following December to May, relative to the detrended autumn SIE in BKS below 0.8 standard deviation (low SON SIE). The dots highlight significant anomalies at a confidence level of 95%.

Atmospheric blocking is a large-scale mid-latitude atmospheric phenomenon primarily associated with persistent anticyclonic quasi-stationary high-pressure systems, which typically extend vertically throughout the troposphere. When the high-latitude jet stream develops large and nearly-stationary meanders (e.g., Fig. 5a), the strong westerly winds cannot flow along their original direction and further blocking occurs. It makes weather systems move more slowly or even become stationary (Rex, 1950), which is particularly important the formation of extreme weather events. In winter, blocking can be responsible for the equatorward advection of polar air masses and the resultant anomalous cold winter temperature (Sillmann and Croci-Maspoli, 2009).

In relation with low SIE, starting in December, a center of enhanced blocking activity over Northern Russia, coupled with a center of relatively weaker blocking activity over Greenland, is observed (Fig. 6a). The blocking activity over Greenland gradually intensifies and moves down the southwest coast of Greenland, reaching its most phase in February (Fig. 6c) before vanishing over time. The center of the blocking activity over Northern Russia remains and subsequently shifts to couple with





a center of enhanced blocking activity over the southeast coast of Greenland (Fig. 6b-c), thus the more robust flow anomaly. Interestingly, there is a persistent blocking at every longitude between the Baffin Bay and Europe from January to February. In February (Fig. 6c), the positive anomalies merge and shift southward, extending more than in January, forming one broad and a continuous intensified block across Greenland and Northern Europe. The configuration of anomalous blocking frequency in March (Fig. 6d) corresponds to Greenland blocking known to be associated with the negative NAO phase (Woollings et al.,

2010), which is usually related to the splitting of the jet stream into two distinct branches (Trigo et al., 2004). In April (Fig. 6e), negative anomalies occupy most of the area with no blocking over Greenland, except for the weak positive in the narrow band in North Atlantic. Then, like other variables, the atmospheric circulation system adjusts to a stable state within a short period.







**Figure 7.** TN10p composite maps for the following December to May, relative to the detrended autumn SIE in BKS below 0.8 standard deviation (low SON SIE). Note that a negative sign is added for better understanding. The dots highlight significant anomalies at a confidence level of 95%.

We examine furthermore the monthly evolution of the spatial structure of extreme cold events in Europe. Figure 7 shows the composite for TN10p anomalies due to the combined effect of the atmospheric circulation variables. The insignificant change in TN10p in December (Fig. 7a) indicates that the cold air from the Arctic has not yet penetrated southwards into mid-latitudes, and the slight warming signals in central Europe may be related to the ridges transporting warm air northward. In January (Fig. 7b), pronounced cold anomalies appear in Europe, especially over Norway and Ireland, while positive anomalies



can be observed over Turkey, as a result from the southerly inflow of warm and humid air masses from the Atlantic (Fig. 5b). The widespread pattern of extreme cold remains robust in February (Fig. 7c) with an amplified magnitude over the central and western Europe, while some regions in the south experienced warmer conditions. In March (Fig. 7d), the cold European anomaly persists and extends to the east, while the warm anomalies in the southern part are expanding in coverage and extend to the northwest, but a significant cooling trend can be seen over the Iberian Peninsula. Even if there is no blocking at the same

time (Fig. 6d), the predominance of these cold anomalies might still be a direct response to be the persistence of blocking situation. In April (Fig. 7e), the TN10p indicate a diploe-like patterns between the eastern and western European, with weaker and incoherent anomalies. The cyclonic circulation around western Europe (Fig. 5f) transports warmer Atlantic air masses to the continent, essentially shaping the general norm encompassing Europe.

        The robust features revealed that persistent winter cold extremes occur in the presence of atmospheric blocking. Overall,

blocking influences the local weather conditions around the block, particularly the European cold extremes are associated with mid- and high-latitude blocking over the European continent as well as the North Atlantic. Europe is the dominant region because of the configuration of a strong and meridionally tilted storm track upstream of a large landmass (Kautz et al., 2022). Northerly advection anomalies mainly cause cold anomalies over most of Europe to the east of the blocking core. Over the North Atlantic, blocking is strongly correlated with the negative phase of the NAO that itself is associated with the

development of cold European winters. During the negative phase of NAO, the synoptic pattern provides diffluent flow conditions favorable for the onset and maintenance of blocking systems (Luo et al., 2015). In addition, blocking occurs over Greenland (e.g., Fig. 6d), which is associated with the negative phase of NAO and has strong downstream impacts in Europe (Rimbu and Lohmann, 2011; Davini et al., 2012; Kretschmer et al., 2018).

### 3.2 Large-scale atmospheric circulation and sea ice variability as captured by climate models

The global climate models serve as crucial tools for simulating the climate variability and change, and numerous studies have assessed climatic extreme events on both global and regional scales (e.g., Screen et al., 2015). To evaluate their performances in the simulation of interannual variability of atmospheric circulation linked to the reduction of sea ice, we repeat the aforementioned analyses using the CMIP6 model simulations (Eyring et al., 2016). Figure 8 presents the changes in autumn SIE over the BKS, and the variability after removing the linear trend. As demonstrated in Fig. 8a, the multi-model mean of

CMIP6 captures a reduction in autumn SIE within the BKS during 1979 to 2021, although this decrease is weaker than observed. However, there exists a substantial divergence between the modeled and the observed SIE variability as illustrated in Fig. 8b. This implies that the resulting composite analyses of atmospheric variables, as shown in Fig. 9, are unrealistic. Few models capture the negative phase of AO and the pattern of a warm Arctic and cold continents, as observed in the reanalysis (Fig. 3) for the low SIE years.






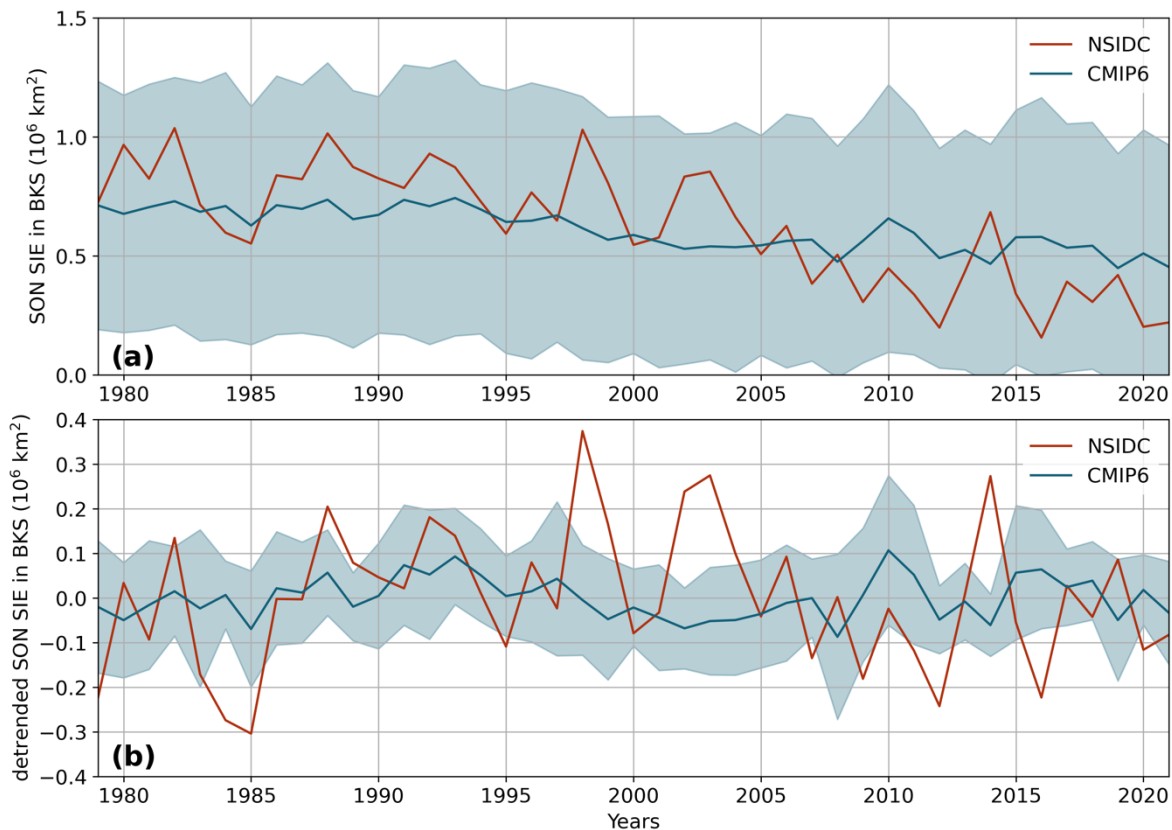

**Figure 8.** Time series of autumn SIE over the BKS from 1979 to 2021 in observations (red) and the CMIP6 multi-model mean (blue), respectively. (a) Original time series and (b) the linear detrended time series. The shaded blue area denotes the range within one standard deviation.


This discrepancy may suggest an inherent limitation of these models to accurately capture the interannual sea ice variability. In addition, recent studies (Anstey et al., 2013; Davini and Cagnazzo, 2014) have shown that some climate models fall short in accurately simulating the physical processes linked to the NAO and the blocking activity over Europe and Greenland. The major implication is that, even though the state-of-the-art climate models improve the simulation of their mean

states, the detailed physics are still of large uncertainty.



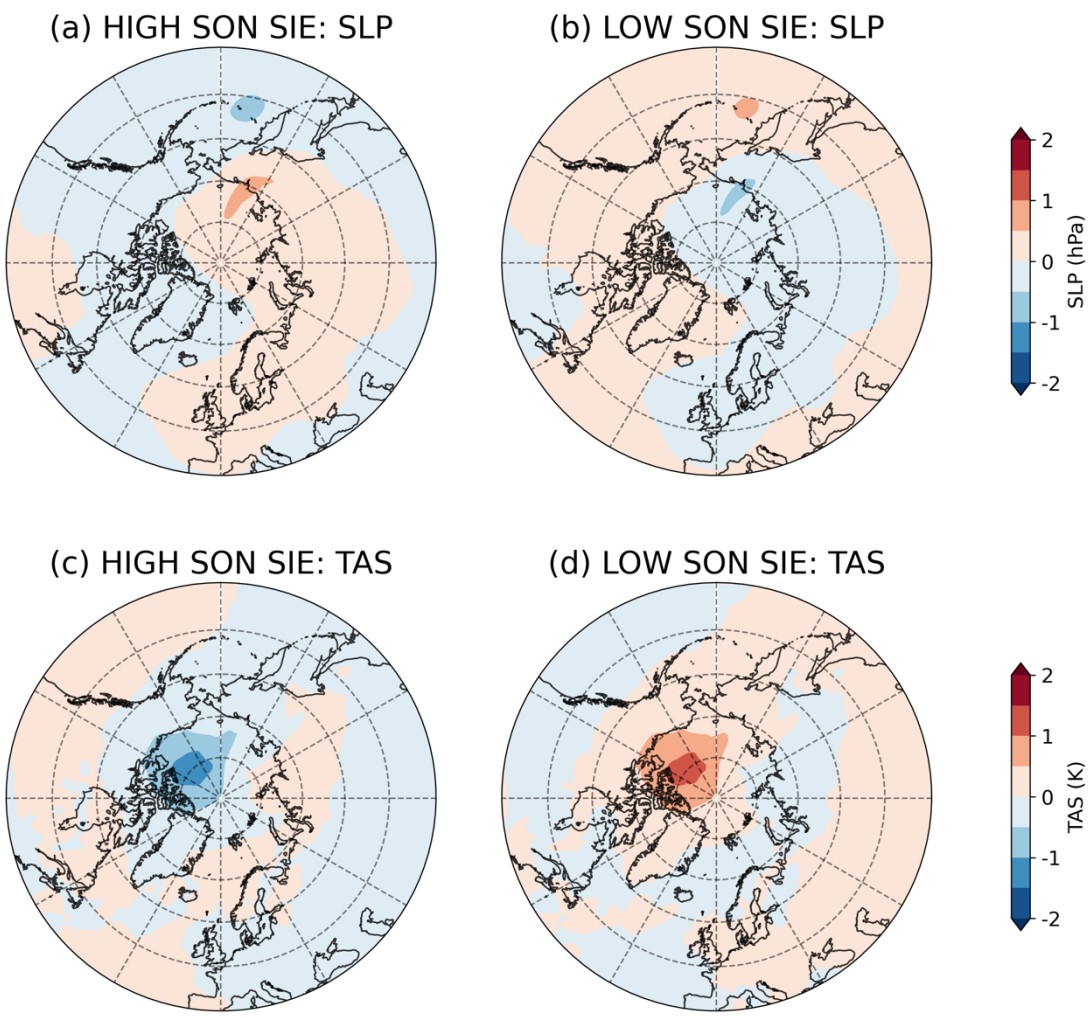

**Figure 9.** Composites of SLP (a, b) and Surface Air Temperature (TAS) (c, d) for winter (DJF) relative to the detrended autumn SIE in the BKS above (left) and below (right) 0.8 standard deviation, as depicted by the CMIP6 multi-model mean results.



**4 Conclusions**


In this study we examine the influence of autumn sea ice loss over the BKS on the severe European extreme winters using observational data to track the interannual variability. We investigate the possible dynamic adjustment pathways of atmospheric circulation and blocking frequency in response to autumn sea ice retreat on a monthly timescale. Our results indicate that diminishing autumn sea ice over the BKS act as a pacemaker for the large-scale atmospheric circulation

rearrangement and the variations in blocking dynamics, steering extreme cold winters over Europe.

The local and remote atmospheric circulation clearly responds to sea ice change over the BKS. The lower troposphere diabatic heat anomaly in the BKS sector associated with reduced sea ice in autumn enhances the upward emanation of the stationary Rossby wave trains. This further perturbs the SPV and triggers the sudden stratospheric warming events in January and February, thus establishing a stratospheric path. When SPV is weaker than usual, the westerly flow is relatively slow and

tends to meander. Additionally, stratospheric warming events can weaken or reverse the stratospheric winds, resulting in circulation features that eventually descend into the troposphere and surface within about one month. This downward influence mechanism explains the deformation in the spatial structure of the tropospheric circulation anomalies and the southward movement of the jet stream. Then, a negative AO/NAO pattern emerges as the dominant tropospheric circulation, which develops in December and reaches its largest amplitude in February.

In addition to changes in the large-scale atmospheric circulation, thermodynamic forcing from reduced sea ice can also contribute to weather conditions through the internal shifts in atmospheric dynamics (Screen, 2017). In particular, blocking activity is critical to European flow changes due to its capability to interfere with the main cyclonic westerly flow, which typically leads to persistent weather conditions around the block for one to two months (Buehler et al., 2011). We demonstrate that cold extremes in Europe can be attributed to the presence and persistence of atmospheric blocking systems over Greenland

and the North Atlantic, especially during late winter (e.g., February). However, the cold anomalies are usually not located directly below the blocking center, but rather downstream or south of it. This is why extreme cold temperatures are most pronounced over central Europe.

The results presented here indicate that not only the atmospheric circulation realignment but also variations in blocking frequency can be crucial for the frequency of occurrence of mid-latitude weather extremes. The negative phase of NAO

provides diffuse flow conditions that further facilitate the onset and sustainment of blocking systems (Luo et al., 2015). The intensified blocking activity and the negative phase of NAO are closely related and work to reinforce each other, each associated with and jointly shaping the spatial distribution of cold anomalies over large parts of European continent (Rimbu et al., 2014; Wegmann et al., 2020). Specifically, cold anomalies at TN10p in large parts of Europe are mainly caused by northerly



flow anomalies in the eastern part of the blocking core region, while warm anomalies in southern Europe correspond to the
cyclonic circulation in the North Atlantic associated with the negative phase of the NAO.

In addition, we observe a significant divergence between the modeled and observed autumn SIE variability, meaning that
the further composite response of the atmospheric conditions to sea ice loss is not accurately represented. This discrepancy
also provides partial explanation for the suboptimal performance of the model climate sensitivity assessments, as these
assessments rely on detailed model physics rather than the mean state performance for evaluating the overall impact on the
climate.

Overall, the Arctic Sea ice does not merely respond passively to global climate change, but instead, its changes affect
weather conditions in other regions bridging spatio-temporal scales (Lohmann et al., 2020). Our results imply that the autumn
anomalous low BKS sea ice in autumn is associated with severe European extremes in subsequent winters. Future work on
this link on a monthly timescale will therefore improve our understanding of the sub-seasonal to seasonal predictability of
midlatitude extreme events, and ultimately advance our projections and predictions of future climate.

*Data Availability.* No datasets were generated during this study. All data used in this study are publicly available from the
sources cited in "Data and Methods".

*Author Contributions.* D.C. and M.I. conceived and designed the study. D.C. carried out the analyses and produced all figures
under the guidance of M.I., G.L., and X.C. And all authors contributed to the rewriting and revising of the manuscript.

*Competing interests.* The authors declare that they have no conflict of interest.

*Acknowledgements.* We acknowledge support by AWI through its research infrastructure in the topic "Ocean and Cryosphere
under climate change" as part of the Helmholtz Program "Changing Earth - Sustaining our Future". Support by the Helmholtz
Climate Initiative REKLIM is gratefully acknowledged. We acknowledge the National Snow and Ice Data Center (NSIDC),
the European Centre of Medium-Range Weather Forecasts (ECMWF), and the Expert Team on Climate Change Detection and
Indices (ETCCDI) for making their data available to us.

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
