# Peer review of "The linkage between autumn Barents-Kara sea ice and European cold winter extremes"

_EGUsphere, 2023_

## Referee Comment (RC1)

**The linkage between autumn Barents-Kara sea ice and European cold winter extremes**

by D. Cai, G. Lohmann, X. Chen, and M. Ionita

This study evaluates the link between sea ice retreat in the Barents-Kara Seas (BKS) and extreme cold winters in Europe using observations of monthly sea ice extent (NSIDC) and ERA5 reanalysis data. Additionally, a set of CMIP6 models is used to assess the performance of climate models in simulating the variability of sea ice and the large-scale atmospheric circulation. The study finds that a reduced sea ice extent in the BKS in autumn is related to cold winters in Europe following the formation of a stationary Rossby wave which facilitates frequent episodes of atmospheric blocking over Greenland and the North Atlantic. The authors further link the increased blocking frequency and negative surface air temperature anomalies over large parts of Europe with the negative phase of the NAO and associate the observed cold anomalies with northerly flow anomalies. Finally, the authors conclude that variability in autumn KBS sea ice extent is not accurately represented in climate models following uncertainty in model physics and that therefore these models cannot be used to correctly illustrate atmospheric variables during low KBS sea ice extent / cold European winters.

The study covers an interesting and relevant topic. As the Arctic is changing rapidly, a better understanding of the linkage between Arctic sea ice loss and weather extremes in Europe is crucial. However, this topic has already been broadly discussed in recent literature and I struggle to find any aspect of this study that is novel. Most of the presented results are already known from previous publications. In addition, there is no consensus about the causality of the discussed link in recent publications, a fact which has not been discussed and questioned adequately in the present study. Finally, the judgement of the authors regarding the ability of climate models to represent BKS sea ice variability is based on the erroneous interpretation of an analysis, causing the authors to draw hasty conclusions about climate model uncertainties. Against this background, not only should the authors question the relevance of their research questions and interpretation of results, but major changes to the manuscript would be needed such that it can contribute to the ongoing scientific discourse about Arctic – mid-latitude connections.

Overall, many profound changes are still required to make this paper potentially suitable for publication. Therefore, I recommend very major changes as outlined in the comments below.

**General comments:**

1. **Novelty of the study**
   The authors point out correctly in their introduction, that there have been many studies discussing a possible linkage between the Arctic and the mid-latitudes. However, some important references and a more detailed discussion, in particular of the controversy regarding the causality and driving mechanisms of such a linkage, are missing. For example, there has been a recent review study by Outten et al. (2022), which covers many aspects of the present manuscript, but additionally includes a detailed review of the different viewpoints on this topic as well as the role of climate models. Outten et al. (2022) concisely summarize the several mechanisms proposed by previous publications that could link Arctic sea ice retreat with cold temperatures in Eurasia. From my point of view, this summary already includes most of the results presented in the conclusion of this manuscript. Furthermore, the observed link to a negative NAO pattern (which is also under debate) has already been established in earlier publications, see for example Honda et al. (2009). Thus, it does not become fully clear to me what the actual "novelty" of this study is.

2. **Causality vs. correlation**
   One aspect which is not adequately discussed by the authors is the importance of a differentiation between causality and correlation when looking at Arctic – mid-latitude linkages. Several studies

emphasize the importance of such a differentiation and in particular advice caution when interpreting a possible causality (e.g., Blackport et al., 2019; Fyfe 2019; Blackport and Screen, 2020). Furthermore, multiple studies based on observations as well as climate model data questioned the causality of a "low sea ice –> cold mid-latitudes" pattern. Instead, they emphasized the role of internal atmospheric variability as driver of circulation changes as opposed to anomalies in Arctic sea ice (e.g., Honda et al., 2009; Kug et al., 2015; Sorokina et al., 2016; Blackport et al., 2019). Thereby these studies argue for example with the sign of the observed heat flux anomalies, which supports the view point that sea ice anomalies are driven by the atmosphere rather than the other way round.

The key question about the mechanism that drives cold European winter temperatures but maybe already the decline in Arctic sea ice in the first place does not seem to be fully answered yet. However, several of these studies indicate that it is actually not sea ice anomalies that drive the atmospheric circulation, but rather preceding anomalies in the atmospheric circulation which then drive both Arctic sea ice loss and cold temperatures in the mid-latitudes, leading to the observed correlation between both. In addition, studies analyzing sea ice loss in multi-model experiments could not find concurrent cold winters which further supports the idea of internal variability as main driving mechanism (e.g., Ogawa et al., 2018; Blackport and Screen, 2021).

A revised version of this manuscript should include a more extensive discussion of this controversy regarding other possible drivers of the observed European winter cold extremes such as internal atmospheric variability (as opposed to sea ice decline as main driver). An additional analysis of atmospheric circulation anomalies preceding the SIE anomalies could help to better understand the causality between the different processes. Without such an analysis, statements such as in L321 "The local and remote atmospheric circulation *clearly* responds to sea ice change" should be avoided.

3. **Robustness of data**
The results of this study are mainly based on nine seasons of observations. With such little available data caution is required when interpreting results as "robust", as the observational record is simply too short, especially when considering the involved internal atmospheric variability. The additional analysis of climate model simulations could strongly improve the robustness of the results.

4. **Analysis of climate model data**
The authors briefly evaluate the representation of BKS sea ice variability in several CMIP6 models and quickly conclude that the modeled SIE variability is much smaller compared to the observed SIE variability, which they attribute to large uncertainty in model physics. Unfortunately, this then leads them to the conclusion that further analyses with the models would lead to unrealistic results. This assessment is based on Fig. 8, where a multi-model mean is used to evaluate the SIE variability in CMIP6 models. However, by taking the multi-model mean, any variability within the models is probably eliminated to a great extent. On contrast, the shading in Fig. 8a is a measure of the interannual variability within the different models. I am not sure why the shading is that strongly reduced in Fig. 8b, when this panel basically shows the same as above using a (linear?) detrending. In my opinion it should be similar in amplitude compared to the shading in panel (a), which would imply a similar variability within the climate models compared to the observational data, as the observations are completely enveloped by the blue shading in panel (a).

Next to a revision of this interpretation of SIE variability, a more detailed analysis of the climate model data is needed. Figure 8a clearly shows a sufficient SIE variability in the CMIP6 models which would allow for a meaningful complementary analysis and discussion of the involved dynamical processes in the climate model data.

5. **Dynamical drivers of the cold temperature anomaly**
   While the authors describe the correlation between the observed circulation patterns and temperature anomalies in detail, the manuscript lacks a more in-depth investigation of the physical processes behind the observed temperature extremes. Are these extremes merely caused by cold air advection? Does radiative cooling further contribute to the cold extremes and if so, how is it linked to, for example, the observed enhanced stationary blocking?

**Specific comments:**

L78ff: How exactly is the data detrended? Is it a linear detrending?

L82: "r1i1p1f1" - I am not sure if everyone knows this notation of climate model ensemble members. Please specify.

L100: What do you mean by "given three reference latitudes"? Does this refer to the latitudes $\phi_0$, $\phi_0+15°$ and $\phi_0-15°$?

Section 2.1: If I am not mistaken, there is no description which area has been defined as the BKS region. It would be very helpful, if for example a map would be added.

L112ff: "…SIE in autumn was above (below) 0.8 standard deviation as high (low) SIE years." I guess the authors mean "above 0.8 or below -0.8 standard deviation"? In L123ff it seems to be correct but then again in the captions of the composite figures the same mistake occurs.

L124ff: I count 9 cases of both high and low autumn SIE episodes (not 8, as stated).

Fig. 1: I assume that the grey-dashed line marks the value of 0.8 standard deviations? If yes, please add this information to the figure caption. Also, the axis label is confusing. The single values of the detrended SON SIE show the anomaly in that particular season. Thus, it would be more intuitive to label the y-axis with "anomaly in SON SIE" or similar.

Fig. 2 and 3: Instead of calling it "Composite maps of XYZ relative to the detrended autumn SIE" it would be easier to understand as "Composite maps of XYZ anomalies during high / low SIE SON".

L148ff: "…, which leads to the increase of SST over the BKS region" – is this statement based on an analysis of correlation/causation between the decrease in sea ice and increase in SST? I assume that also a positive SST anomaly could lead to the observed SIE decrease. Maybe also add a sentence, how an increase in SST strengthens the Siberian High in winter.

Fig. 6: Be more precise with the colorbar label. Instead of "Block" I would label it as "blocking frequency".

L246: I do not agree that there is persistent blocking between Baffin Bay and Europe in January and February. Figure 6 shows a monthly mean blocking frequency of approximately 10% in this area. This does (1) not mean that there is persistent blocking in both months (but only in 10% of the time steps) and (2) this statement implies that the whole region is affected by blocking at the same time which is not necessarily true (maybe check daily composites of blocking frequency for a more refined interpretation).

Fig. 7: "Note that a sign is added for better understanding" – I do not understand this note, please refine.

Fig. 8: What does the range in panel (b) show? As it is much more reduced compared to the range shown in panel (a) this is surely not the 1σ-range of the different model runs?

L292 and Fig. 9: How many high/low SIE seasons are included in these composites?

**Technical corrections:**

Since the manuscript needs some thorough revision, I only add some more general corrections here.

L30 and elsewhere: Arctic Sea ice → Arctic sea ice
Consistency: either always "sea ice conditions" or "sea-ice conditions"
L38: McCusker et al., 2016; Sun et al., 2016)
L125 and elsewhere: following the WCD rules for figure references: if not at the beginning of a sentence, then "Figure 1" → "Fig. 1"

**References:**

Blackport, R., and Screen, J. A. (2020): Insignificant effect of Arctic amplification on the amplitude of midlatitude atmospheric waves. Sci. Adv., 6, eaay2880. doi:10.1126/sciadv.aay2880.

Blackport, R., and Screen, J. A. (2021): Observed statistical connections overestimate the causal effects of Arctic sea ice changes on midlatitude winter climate. J. Clim., 34, 3021-3038. doi:10.1175/JCLI-D-20-0293.1.

Fyfe, J. C. (2019). Midlatitudes unaffected by sea ice loss. Nature Clim. Change, 9, 649–650. doi:10.1038/s41558-019-0560-3.

Kug, J.-S., Jeong, J.-H., Jang, Y.-S., Kim, B. M., Folland, C., Min, S.-K., and Son, S.-W. (2015). Two distinct influences of Arctic warming on cold winters over North America and East Asia. Nature Geosci., 8, 759-762. doi:10.1038/NGEO2517.

Ogawa, F., Keenlyside, N., Gao, Y., Koenigk, T., Yang, S., Suo, L., Wang, T., Gastineau, G., Nakamura, T., Cheung, H. N., Omrani, N.-E., Ukita, J., and Semenov, V. (2018): Evaluating impacts of recent Arctic sea ice loss on the Northern Hemisphere winter climate change. Geophys. Res. Lett., 45, 3255-3263. doi:10.1002/2017GL076502

Outten, S., Li, C., King, M. P., Suo, L., Siew, P. Y. F., Cheung, H., Davy, R., Dunn-Sigouin, E., Furevik, T., He, S., Madonna, E., Sobolowski, S., Spengler, T., and Woollings, T. (2022). Reconciling conflicting evidence for the cause of the observed early 21st century Eurasian cooling. Weather Clim. Dynam., 4, 95-114. doi:10.5194/wcd-4-95-2023.

Sorokina, S. A., Li, C., Wettstein, J. J., and Kvamstø, N. G. (2016): Observed atmospheric coupling between Barents Sea ice and the warm-Arctic cold-Siberian anomaly pattern. J. Clim., 29, 495-511. doi:10.1175/JCLI-D-15-0046.1.